# Association between Precarious Employment and Chronic Stress: Effect of Gender, Stress Measurement and Precariousness Dimensions—A Cross-Sectional Study

**DOI:** 10.3390/ijerph19159099

**Published:** 2022-07-26

**Authors:** Mireia Julià, Fabrizio Méndez-Rivero, Álex Gómez-Gómez, Óscar J. Pozo, Mireia Bolíbar

**Affiliations:** 1Research Group on Health Inequalities, Environment, and Employment Conditions (GREDS-EMCONET), Department of Political and Social Sciences, Universitat Pompeu Fabra, 08005 Barcelona, Spain; mireia.bolibar@uab.cat; 2ESIMar (Mar Nursing School), Parc de Salut Mar, Universitat Pompeu Fabra-Affiliated, 08003 Barcelona, Spain; 3SDHEd (Social Determinants and Health Education Research Group), IMIM (Hospital del Mar Medical Research Institute), 08003 Barcelona, Spain; 4Applied Metabolomics Research Group, IMIM (Hospital del Mar Medical Research Institute), 08003 Barcelona, Spain; agomez@imim.es (Á.G.-G.); opozo@imim.es (Ó.J.P.); 5Centre d’Estudis Sociològics Sobre la Vida Quotidiana i el Treball, Institut d’Estudis del Treball, Universitat Autònoma de Barcelona, 08193 Cerdanyola del Vallès, Spain

**Keywords:** cortisol, chronic stress, precarious employment, gender, social determinants of health

## Abstract

Precarious employment has been highlighted as a social determinant of health, given, among others, to its alleged association with chronic stress. However, few studies have been conducted analyzing such association, using both perceived stress indicators and biological markers. Accordingly, the present study analyzed the association of multidimensional (6 dimensions) precarious employment scale with perceived stress and 23 markers of adrenal and gonadal hormone production, including cortisol. The sample consisted of 255 salaried workers from Barcelona (125 men, 130 women) aged 25–60. OLS regression models stratified by sex were conducted. Results demonstrated that precarious employment increased the probabilities of having perceived stress in both sexes. In addition, the production of adrenal hormones among men is associated with precarious wages and among women with precarious contracts (“Temporariness”, “Disempowerment”, and “Rights” dimensions). Therefore, precarious employment could be embodied by workers, altering their perceived well-being and physiological characteristics. Differences between men and women in the physiological effect of precarious employment could express not just the biochemical differences inherent to biological sex, but also the social construction of gender identities, positions and roles in society and family, as well as gender inequalities in the labour market.

## 1. Introduction

Precarious employment is considered a social determinant of health and of health inequalities due to the importance of work in people’s lives [1]. In recent decades and especially in recent years, the economic crisis, technological and political changes, and, more recently, the COVID-19 pandemic, have increased the precariousness of employment conditions due to the greater flexibility of the labour relations conducted by the flexibilisation of labour regulations [2,3,4]. Although there is no standard and consensual definition of precarious employment, different approaches exist to measure it [5]. One of the most used is based on single-dimensional measurements, that is, from a single variable such as temporariness [6,7,8] or perceived insecurity [9,10,11]. In recent years, however, different multidimensional approaches have emerged [5,12,13] because precarious employment has been observed as a widespread phenomenon not only related to the type of contract [14]—since employees with permanent contracts can also find themselves in a precarious situation—but also related to other characteristics of labour relations such as wages, the power to exercise labour rights, helplessness in the face of authoritarian treatment, or the level of negotiation of employment conditions. 

One of the known consequences of precarious employment is the negative influence on the working population’s health. In recent decades, many studies have been conducted analysing the impact of precarious employment on both physical and mental health [8,15,16,17,18]. Indeed, the results demonstrated that employees in highly precarious employment measured multidimensionally had worse mental and/or self-perceived health than employees in less precarious employment [12,14,19,20,21,22]. Various studies also demonstrate that there is more precarious employment among certain groups of workers such as women, young people, immigrants, and manual workers [23,24,25]. In particular, women’s insertion in the labour market is more precarious as a result of persistent vertical and horizontal occupational gender segregation, whereby women get access to less prestigious occupations and within them to less important positions [26], which are usually subject to worse working conditions [27]. Although several studies have indicated that women’s greater exposure to precarious employment increases the likelihood of poorer physical and mental health [28,29], little is known about the psychophysiological response of men and women to overall precarious employment or its different dimensions and their relationship with gender stereotypes. Thus, for example, the literature demonstrates that low wages or income can cause stress through increased precariousness and eventually lead to mental health problems [30,31,32]. Still, little is known about whether there are gender differences in the stress impact of these dimensions or whether any of the dimensions has greater weight than others in this impact on health. Hence it is important to use multidimensional measures of precarious employment, not only through global indices but also through the sub-dimensions that compose them. 

Stress is a major problem in today’s society, with significant effects at different levels. Studies demonstrate how “stressful events” are associated with poor mental and physical health through psychophysiological mechanisms [33] and can eventually lead to different health problems such as cardiovascular disease, metabolic syndrome, osteoporosis and/or depression [34]. As far as we are aware, research into the relationship between precarious employment and stress is scarce, and the existing research only uses single-dimensional measures such as temporariness [35] or perceived insecurity [36]. However, no studies analyse multidimensional precarious employment and stress, other than the studies related to the PRESSED project [37,38] which include this study. 

In addition, in this field of social epidemiology, stress is studied mainly through self-reported measures such as the Perceived Stress Scale (PSS) [39]. The literature demonstrates that a relationship exists between presenting high levels of stress measured using the PSS and poor health. Thus, it has been observed that in patients who have suffered an acute myocardial infarction and have a moderate/high level of stress measured using the PSS had increased 2-year mortality and have a poor 1-year health status compared to those who have low levels of stress [40]. Additionally, patients with peripheral artery disease who suffer from chronic stress after six months of diagnosis have higher odds of worse recovery at 12 months [41]. Chronic stress has also been related to nutritional problems such as emotional eating [42].

Even scarcer in this field is the use of objective measures such as biomarkers. Biomarkers should theoretically allow greater comparability between cases since they should not be so affected by biases in the subjective interpretation of situations. One of the most used biomarkers for studying stress is cortisol. Cortisol is a glucocorticoid steroid hormone that is part of the hypothalamic–pituitary–adrenal (HPA) axis, involved in the body’s response to stress by mobilizing resources to supply energy. It also participates in the regulation of other important systems such as the immune system, the cardiovascular system or in affective and cognitive processes [43]. Cortisol can be measured in serum, saliva, or urine, but these determinations may be affected by different factors such as circadian rhythm or needle apprehension. To overcome these limitations, the determination of cortisol in hair is increasingly used in the evaluation of chronic stress [44]. However, different studies have demonstrated conflicting results in the association between hair cortisol levels and subjective stress measures [45,46]. These divergences are also found in the association between hair cortisol levels and some stressful situations or contexts. Thus, some studies demonstrate that there is an increase in cortisol levels in the hair in situations such as unemployment, shift work, chronic pain, or major life events [47]. At the same time, cortisol levels in hair have also been reported to increase in cases diagnosed with depression, but decrease in disorders such as anxiety [47]. Due to these divergent results, in recent years, some authors have questioned the direct relationship between hair cortisol concentration and different measures of self-reported chronic stress, cast doubt on the suitability of cortisol as the best biomarker to study chronic stress [46,48,49,50,51,52]. Thus, other steroid biomarkers related to the HPA axis (e.g., cortisol metabolites) [53] or to other axes such as the hypothalamic–pituitary–gonadal (HPG) axis have been recently used. For example, testosterone or dihydroepiandrosterone [54,55] may provide new insights into the relationship between steroidal hormones, stress, and precarious employment.

Against this background, the objective of this study is to examine whether there are differences in the association between multidimensional precarious employment and stress measured through subjective and self-reported measures or using biomarkers of the HPA/HPG axes and whether there are gender differences in these associations.

## 2. Materials and Methods

### 2.1. Study Design and Sampling

The cross-sectional study is based on a sample of 255 employees from Barcelona, Spain, aged between 25 and 60 years (125 men and 130 women). The sample is non-probabilistic based on proportional quotas by sex, age group (25–34 years vs. 35–60 years), place of origin (born in Spain vs. born abroad) and the socioeconomic level of the district of residence (middle, upper-middle or high vs. low-middle- and low-income districts). Participants were selected from the records of the 2017 Barcelona Health Survey, from which 215 cases were obtained of persons who agreed to participate in the study. In addition, 40 subjects were recruited through social and workers’ organisations in order to compensate for the bias of the initial sample towards profiles of higher levels of education and income. In all cases, the inclusion criteria were: (i) being a salaried worker or a bogus self-employed worker (an independent worker who reports self-employment income but serve a single employer, concealing a dependent employment relationship), (ii) being between 24 and 60 years old, (iii) living independently in Barcelona (i.e., persons living with their parents were excluded), (iv) the length of hair at the back of the head being of at least one centimetre, and (v) not having taken holidays or leave from work within the month prior to the interview. The exclusion criteria were: (i) having taken corticosteroids within the month prior to the interview, (ii) having been diagnosed with an adrenal disease, and (iii) being pregnant, due to possible alterations in cortisol levels as a result of gestation. 

Each sample subject attended a face-to-face interview of approximately 40 min in which a questionnaire was administered that included items on the different subjects of interest of the study (precarious employment, working conditions, uncertainty, support networks, perceived stress, and physical and mental health), as well as questions on sociodemographic characteristics.

Each participant was also asked for a hair sample to be able to analyse cortisol and other metabolites, used as potential biomarkers of chronic stress. The first centimetre of the lock of hair that is in contact with the scalp is the biological material subjected to laboratory analysis using a previously validated method known as liquid chromatography-tandem mass spectrometry (LC-MS/MS) [56]. One centimetre of hair was collected in order to obtain information concerning the steroids produced during the month prior to sample collection.

More detailed information on the project can be found in the protocol [38] as well as concerning the EPRES scale, its dimensions, and items [57]. 

### 2.2. Variables 

As outcomes, two groups of variables were taken: the Perceived Stress Scale (PSS) and the set of biomarkers analysed. On the one hand, perceived stress was measured using the Spanish version 2.0 of the Perceived Stress Scale (PSS), based on the complete 14-item Likert-type version by Cohen [39], adapted by Sanz-Carrillo [58]. The raw score is calculated by summing the figures for the 14 items, which range from 0 to 4. The minimum value observed in the sample was 1 and the maximum 44. 

Moreover, a set of adrenal and gonadal steroids and metabolites was measured in hair. The adrenal ones provide information on the HPA axis, and include the levels of cortisol, 20α-dihydrocortisol (20αDHF), 20ß-dihydrocortisol (20βDHF), cortisone, 20α-dihydrocortisone (20αDHE), 20ß-dihydrocortisone (20βDHE), cortolone, 11-dehydrocorticosterone and androstenedione (AED). The gonadal ones provide information on the HPG axis and include levels of testosterone and progesterone as well as AED. In addition to the capillary concentrations of the selected steroids, several ratios were included to evaluate the activity of key enzymes in steroid production and metabolism. For example, the cortisol/cortisone ratio was calculated to evaluate the activity of the enzyme 11ß-hydroxysteroid dehydrogenase (responsible for cortisol to cortisone interconversion). Moreover, the relative abundance of each steroid (as a %) was calculated as an additional marker, to account of the amount of cortisol relative to its metabolites. This minimises the possible effect of different steroid adsorption in hair depending on hair type.

Precarious employment was used as an explanatory variable. This was measured using the Employment Precariousness Scale (EPRES), an instrument that has been validated in various countries [19,23,25,59,60]. The scale consists of 22 indicators classified into six dimensions: temporariness—duration of the contractual relationship, wages—the ability of the employment to generate income to cover needs, labour rights—social coverage associated with employment, the exercise of rights—the ability, in practice, to exercise labour rights, vulnerability—level of helplessness in the face of authoritarian treatment, and disempowerment—capacity for collective bargaining and influence on employment conditions [57]. Each dimension contributed equally to the total score, regardless of its number of items. To obtain a scale of equal weight, the score of each dimension was calculated independently, standardised, and integrated into an overall score that results from the sum of the items of each dimension. Thus, each dimension is a subscale whose score was transformed to range between 0 and 4. Finally, the overall EPRES score was calculated, ranging from 0 (not at all precarious) to 4 (highly precarious), based on the mean of all scales [23]. For this study, the current parameters in the wages dimension were updated. Exploratory, confirmatory factor analyses and Cronbach’s alpha coefficients were used to evaluate the scales’ validity and reliability, respectively (Appendix A). Regarding the validity, factor-loading estimates revealed that all items were related to their theorised dimensions.

Regressions with biomarkers were adjusted for age and body mass index (BMI) since weight can partially determine these biological outcomes [46]. Regressions with PSS were adjusted only for age.

### 2.3. Statistical Analysis 

Initially, a description of the studied sample was made, for which the means and their standard deviations were calculated. The variables corresponding to the biomarkers were transformed to a logarithmic scale to normalise their distribution. 

A linear regression model was used to estimate the association between precarious employment and the PSS adjusted for age. Subsequently, similar models were carried out for each of the biomarkers studied, but in this case, in addition to adjusting for age, they were also adjusted for BMI. These same analyses were repeated for each of the dimensions that comprise the EPRES. All analyses were stratified by sex and performed using Stata 16.0.

## 3. Results

### 3.1. Descriptive Results

The characteristics of the study sample (*n* = 255) are shown in Table 1. The mean age for men was 41.68 years and for women 42.75 years, but these differences were not significant. On average, men had a BMI of 25.34 and women 24.75 and neither were significant. Regarding precarious employment, differences were found between men and women for the EPRES wages dimension, revealing that women have higher levels of precarious employment in this dimension (mean 1.44 vs. 1.13, *p* = 0.013). In the other dimensions, however, no differences were found. Among women, the level of perceived stress (PSS) was significantly higher than among men (mean = 26.12 vs. 22.50, *p* < 0.001). Regarding markers, significant differences were found between the two sexes for 20βDHF, 20βDHE, cortisone, cortolone, 11-dehydrocorticosterone, AED, %20βDHE, which had higher levels among men; and markers %20αDHF and %20αDHE and ratios 20αDHF/20βDHF and 20αDHE/20βDHE, which were higher among women.

### 3.2. Precarious Employment, Perceived Stress, and Production of Adrenal and Gonadal Steroids

Table 2 presents the association of precarious employment with the PSS adjusted for age and with the markers adjusted for age and BMI among men and women. The Appendix A, shows the association of precarious employment with the markers adjusted only for age and stratified by sex.

Precarious employment was positively associated with the PSS both in men (β = 4.82; 95% CI: 2.56–7.08) and in women (β = 5.49; 95% CI: 3.35–7.63). However, among men, no associations were found between the global scale of precarious employment and biomarkers. Among women, meanwhile, the scale of precarious employment was associated positively with 20βDHE, %20βDHE, 20βDHF/cortisol, 11-dehydrocorticosterone and negatively with cortisol/cortisone and %cortisol. 

### 3.3. Perceived Stress and Production of Adrenal and Gonadal Steroids: Their Association with the Dimensions of Precarious Employment

Table 3 shows the association between each of the dimensions of precarious employment and the PSS and the markers. Only the results of the steroids that were significantly associated (*p* < 0.05) with any of the dimensions of precarious employment are presented. 

The “Vulnerability” dimension was associated with the PSS both in men (β = 1.78; 95% CI: 0.48–3.08) and in women (β = 2.07; 95% CI: 0.80–3.34). In addition, among men the “Wages” dimension was also associated with the PSS (β = 0.19; 95% CI: 0.04–0.35). No significant association was found between the other dimensions and the PSS.

Among men, the wages dimension was the one associated with more fundamental adrenal markers (cortisol; 20αDHF; 20βDHF; 20αDHE; 20βDHE; cortolone; 11-dehydrocorticosterone; %cortisone). There was also an association between the dimension of “Disempowerment” with Testosterone and “Exercise of rights” with AED, both of gonadal origin. Furthermore, it was observed that among women, the dimensions of precarious employment with which most biomarkers were associated were “Temporariness” (cortisol, 20βDHF; 20αDHE; cortolone; %20βDHF); and “Disempowerment” (E/A; cortisol/cortisone; %cortisol; 20αDHF/cortisol; 20βDHF/cortisol). However, associations were also found for other dimensions such as “Wages” (20αDHE/20βDHE; %20βDHF; %20αDHE); “Rights” (cortolone; 20αDHE/20βDHE), and “Exercise of rights” (cortisol/cortisone). All these markers correspond to the adrenal axis. The “Vulnerability” dimension was not associated with any of the markers evaluated. As expected from Table 2, women present the highest number of associations between the EPRES dimensions and the markers (five out of six).

## 4. Discussion

The main objective of this article was to examine the association between multidimensional precarious employment and chronic stress measured both through subjective and self-reported measures and using steroid biomarkers, and to analyze whether there are differences between men and women in these associations. To measure precarious employment, a multidimensional global score was considered for the six dimensions together (temporariness, wages, labour rights, exercise of rights, vulnerability, and disempowerment), and a measure for each of the dimensions separately. For the overall score, the results found suggest that precarious employment could be a risk factor for chronic stress both at perceived and hormonal levels. 

At the level of perceived stress, a robust positive association has been found between precarious employment and PSS results in both sexes. At the hormonal level, positive associations have been found between precarious employment and several markers related with cortisol metabolism especially among women, suggesting that the overall score of precarious employment is associated with an increase in cortisol metabolism. In fact, both the amount of cortisol relative to its metabolites (%cortisol) and the amount of cortisol relative to its inactive form (cortisol/cortisone) are negatively associated with precarious employment (being significant in women) while some ratios indicative of cortisol metabolism (20αDHF/cortisol, 20βDHF/cortisol) are positively associated. These results suggest that cortisol overproduction could lead to an increase in its metabolism (in order to minimise exposure to cortisol) and be reflected in the hair due to the accumulation of some of its metabolites and not necessarily the accumulation of cortisol itself.

The association between precarious employment and the PSS is more consistent in terms of statistical significance than the association between precarious employment and biological markers. This result does not permit a single interpretation, as there are several factors that could explain this. Some potential interpretations are that: (i) experiencing stressful phenomena and the self-perception of chronic stress is not necessarily associated with a chronic physiological increase in adrenal glucocorticoid production; (ii) the accumulation of cortisol and other adrenal steroids in the hair may be altered by various factors such as hair type or age [61]; (iii) biological markers are less sensitive to small changes in precarious employment and require greater variations to be modified and display significant differences; and (iv) the two measures provide additional information of different dimensions of precariousness. 

Another of the main findings of this study are the gender differences found in the relationship between the dimensions of precarious employment and chronic stress. The analysis of each dimension of precarious employment separately allows connecting the differences between men and women in the physiological response to precarious employment with the social construct of gender roles that is not observed in the general index. In other words, these differences could demonstrate a complex process of the embodiment of gender roles, in which the biochemical differences inherent in biological sex are related with gender stereotypes and their relationship with the insertion of individuals in the employment market. 

The results obtained seem to indicate that the dimension that most influences the results of perceived stress (measured using the PSS) is vulnerability, which refers to helplessness against authoritarian treatment by the company. Interestingly, this dimension shows no association with any of the steroidal markers considered, either in men or women. At the same time, the dimensions that were associated with the markers did not have associations with the PSS (except the dimension of wages, in men). This result could indicate that the PSS and biomarkers provide complementary information on psychophysiological response to precarious employment. That is, they are not substitute measures that can be used alternatively as indicators of stress associated with precarious employment, indicating that stress at the perceived level does not necessarily have its correlation at the hormonal level. These results seem to indicate that the physiological processes whereby precarious employment generates hormonal responses are not, in themselves, conscious, but that precarious employment can be embodied independently of the consciousness of individuals [62].

From a biochemical standpoint, the results of this study suggest the existence of a gender bias in the stressors causing the overactivation of the HPA axis. Thus, in women there is a clear association between temporariness and the activation of the HPA axis (shown by a positive association not only with the amount of cortisol but also with that of several of its metabolites) showing that temporariness, and therefore its associated uncertainty, affect them more. Other markers related to cortisol metabolism (20αDHF/cortisol, 20βDHF/cortisol, cortisol/cortisone) are associated with dimensions such as “Disempowerment” or “Exercise rights”. These dimensions mainly refer to the contractual dimension of precarious employment, which indicates that women are mainly affected by the characteristics of labour relations. This result acquires meaning in the context of gender inequalities in the labour market in variables such as flexibility, temporariness, and underemployment, among others [27,63,64]. Persistent occupational gender segregation is a structural problem that is directly linked to the precarious insertion of women into the employment market, at both the level of occupations (horizontal segregation) and of the positions they hold within them (vertical segregation) [26,27].

Regarding men, the results of this study demonstrate that the overactivation of the HPA axis occurs basically with precarious wages. This activation is highlighted not only by the association between wages and cortisol but also by the positive association with several of its metabolites (20αDHF, 20βDHF, 20αDHE, 20βDHE and cortolone). In other words, men are mainly affected by precarious wages, a dimension closely linked to the stereotype of the “male breadwinner”. It refers to the traditional model of articulation of employment and family life according to which the male household head acts as the main provider for his dependent wife and children [65]. Despite being a declining work-family arrangement in Spain, it still persists as a powerful social imaginary, especially among men [66]. Although it should be deepened in future studies, this result is in line with the literature in the field of gender roles and stress. Since a classic study pointed out that masculine gender role socialisation affects whether men evaluate certain situations as stressful or not [67], several subsequent investigations have agreed that men have a greater tendency than women to feel emotionally affected if they receive insufficient wages [68].

In this study, chronic stress has been conceptualised as a risk factor for both physical and mental health and not exclusively as a mental health outcome. The association of the EPRES with the PSS and biological markers of stress affirms that precarious employment is a prominent social determinant of health, as has been indicated in the literature [1]. Our results align with previous studies that found associations between multidimensional precarious employment and several physical and mental health outcomes [12,14,19,20,21,22]. They are also consistent with studies based on single-dimensional measures of precarious employment such as temporariness [35] or perceived insecurity [36]. In turn, the hypothesis of multidimensional precarious employment as a possible stressor was already assessed in a previous study related to the PRESSED project (which includes this study), with similar results [37].

Finally, the results of this research could have important implications in the policy and regulation levels. Interventions and policies to deal with work-related stress are normally focused on individual behaviours, physical risks and psychosocial risk factors resulting from work organisation. However, our results indicate that precarious employment conditions can also be a risk factor for work-related stress. Thus, precarious employment is reaffirmed as a prominent social determinant of health that could be faced by public policies in at least three possible ways: first, regularly measuring the magnitude of the phenomenon, which implies its inclusion within the government official statistics. Second, promoting that companies’ OHS practices take into account employment conditions and the characteristics of the labour relation and not only focus on physical and psychosocial working conditions. Third, the government should act on the root causes of precarious employment, implementing policies that improve employment conditions, such as ensuring salary levels, guaranteeing labour rights and their exercise, reduce temporality and job insecurity, among others. On the other hand, our results suggest that precarious employment does not impact the health of men and women in the same way, which alerts about the importance of generating policies that reverse gender occupational segregation in the labour market.

Future studies should delve into the gender-related pathways linking precarious employment with chronic stress.

### Limitations and Strengths

As this is a cross-sectional design, it is not possible to establish a direct causal relationship between steroid production and precarious employment. In addition, there is no information on the period during which subjects have been exposed to precarious employment, which could alter the results. Longitudinal studies could overcome these limitations. Another limitation of the study is that it refers only to the wage-earning population, and it excludes a portion of non-standard work, such as self-employment or informal labour, etc.

On the other hand, one notable strength of this article is its use of biological markers, which is novel not only in the study of precarious employment but also in the field of social epidemiology in general, where subjective and/or self-reported health measures are usually used. In this sense, the interdisciplinary approach of this study is noteworthy, since by combining social and biochemical aspects, it provides novel evidence on the psychophysiological response of multidimensional precarious employment. In turn, the analysis of the different dimensions of precarious employment separately revealed that this response is substantially different for men and women. Moreover, the simultaneous study of the two axes, gonadal and adrenal, by the determination of both hormones and metabolites in hair, is a novelty in biochemical research, given that in previous studies only steroids of the adrenal axis have been studied.

## 5. Conclusions

The results of this study demonstrated that multidimensional precarious employment was associated with chronic stress, although the association with perceived stress was more robust than with biological markers. However, when breaking down precarious employment into its dimensions, differences were observed in the association with the PSS and biological markers, which suggest that both measures provide complementary information on the psychophysiological response to precarious employment. In turn, a gender bias was observed in the overactivation of the HPA axis, which must be understood in the articulation of the biochemical differences inherent to biological sex, with gender socialisation and its relationship with gender inequalities in the insertion of individuals in the employment market.

## Figures and Tables

**Table 1 ijerph-19-09099-t001:** Characteristics of the study population stratified by sex. Precarious Employment and Stress Study sample, 2020. [SD = Standard deviation].

	Men (*n* = 125)		Women (*n* = 130)		*p*-Value
	Mean	SD ^1^	Mean	SD ^1^	
Age (years)	41.68	9.84	42.75	9.79	0.383
Body mass index (BMI) (kg/m^2^)	25.34	3.59	24.75	4.31	0.233
Precarious employment (EPRES)					
Global scale	1.04	0.56	1.02	0.55	0.797
Temporarines	1.27	1.20	1.04	1.13	0.124
Wages	1.13	0.93	1.44	1.06	0.013
Rights	0.45	0.76	0.38	0.64	0.445
Disempowerment	1.40	1.16	1.26	1.16	0.341
Excercise of rights	0.62	0.77	0.69	0.82	0.523
Vulnerability	1.18	0.98	1.12	0.96	0.625
Perceived Stress Scale (PSS)	22.50	7.59	26.12	7.43	0.000
Adrenal and gonadal steroids (ng/mg)					
Cortisol	12.42	18.90	9.33	7.35	0.090
20αDHF	0.99	1.15	0.99	0.96	0.977
20βDHF	5.67	4.19	4.46	2.71	0.007
20αDHE	10.20	8.42	9.05	6.41	0.223
20βDHE	7.48	5.70	5.37	3.26	0.000
Cortisone	33.89	18.37	26.88	17.20	0.002
Cortolone	8.73	3.94	7.09	3.08	0.000
11-Dehidrocorticosterone (A)	2.99	1.59	2.54	1.57	0.023
Testosterone	2.07	2.10	3.08	24.36	0.638
Androstenedione (AED)	5.43	3.07	3.91	2.90	0.000
Progesterone	232.62	1511.24	27.01	32.96	0.130
20αDHF/20βDHF	0.17	0.17	0.20	0.10	0.048
20αDHE/20βDHE	1.38	0.33	1.65	0.39	0.000
Cortisone/11-Dehydrocorticosterone (E/A)	13.01	7.09	12.97	8.82	0.968
Cortisol/Cortisone	0.36	0.45	0.36	0.24	0.901
%Cortisol	15.12	9.25	16.04	6.42	0.359
%Cortisone	50.54	10.29	48.51	9.16	0.097
%20αDHF	1.23	0.76	1.61	0.82	0.000
%20βDHF	8.18	2.74	8.24	2.15	0.843
%20αDHE	14.30	3.92	15.85	4.39	0.003
%20βDHE	10.63	2.80	9.75	2.18	0.006
20αDHF/Cortisol	0.09	0.06	0.11	0.08	0.030
20βDHF/Cortisol	0.65	0.31	0.59	0.28	0.074

^1^ SD = Standard deviation.

**Table 2 ijerph-19-09099-t002:** Linear regression coefficients and 95% confidence intervals (CI) for EPRES global scale predicting Production of Adrenal and Gonadal steroids and PSS, adjusted for control variables and stratified by sex. Precarious Employment and Stress Study sample, 2020.

Outcome ^1^	Men (*n* = 125)		Women (*n* = 130)	
	β	95% CI	β	95% CI
Perceived Stress Scale	4.82 **	(2.56–7.08)	5.49 **	(3.35–7.63)
Adrenal and gonadal steroids				
Cortisol	−0.00	(−0.27–0.27)	−0.05	(−0.27–0.17)
20αDHF	0.08	(−0.25–0.42)	0.11	(−0.15–0.37)
20βDHF	0.06	(−0.14–0.26)	0.11	(−0.05–0.27)
20αDHE	0.04	(−0.15–0.24)	0.18	(−0.00–0.35)
20βDHE	0.05	(−0.14–0.25)	0.18 *	(0.02–0.34)
Cortisone	0.04	(−0.14–0.22)	0.12	(−0.05–0.29)
Cortolone	0.02	(−0.16–0.19)	0.01	(−0.16–0.17)
11-dehydrocorticosterone (A)	0.13	(−0.04–0.29)	0.18 *	(0.01–0.36)
Testosterone	0.16	(−0.06–0.38)	0.08	(−0.28–0.45)
Androstenedione (AED)	0.09	(−0.08–0.27)	0.16	(−0.04–0.37)
Progesterone	0.14	(−0.53–0.80)	0.12	(−0.43–0.67)
20αDHF/20βDHF	0.01	(−0.22–0.24)	0.00	(−0.16–0.16)
20αDHE/20βDHE	−0.01	(−0.08–0.06)	−0.00	(−0.07–0.06)
Cortisone/11-dehydrocorticosterone (E/A)	−0.08	(−0.26–0.10)	−0.07	(−0.27–0.14)
Cortisol/Cortisone	−0.05	(−0.25–0.16)	−0.17 *	(−0.33–−0.00)
%Cortisol	−0.04	(−0.18–0.10)	−0.16 **	(−0.28–−0.05)
%Cortisone	0.01	(−0.07–0.09)	0.01	(−0.06–0.08)
%20αDHF	0.04	(−0.19–0.28)	0.00	(−0.17–0.17)
%20βDHF	0.02	(−0.08–0.13)	−0.00	(−0.08–0.08)
%20αDHE	0.01	(−0.07–0.09)	0.07	(−0.02–0.15)
%20βDHE	0.01	(−0.07–0.10)	0.07 *	(0.00–0.14)
20αDHF/Cortisol	0.09	(−0.16–0.34)	0.16	(−0.02–0.34)
20βDHF/Cortisol	0.06	(−0.13–0.25)	0.16 *	(0.02–0.30)

^1^ All outcomes have been transformed into logarithms to correct skewness. ** *p* < 0.01, * *p* < 0.05.

**Table 3 ijerph-19-09099-t003:** Linear regression coefficients and 95% confidence intervals (CI) for Production of Adrenal and Gonadal steroids and EPRES dimensions (Temporariness, Vulnerability, Wages, Rights, Disempowerment and Exercise of rights), adjusted for control variables and stratified by sex. Precarious Employment and Stress Study sample, 2020.

	Temporariness		Vulnerability		Wages		Rights		Disempowerment		Exercise Rights	
3a. Men. N = 125	β	95% CI	β	95% CI	β	95% CI	β	95% CI	β	95% CI	β	95% CI
PSS	−0.04	(−0.17–0.09)	1.78 **	(0.48–3.08)	0.19 *	(0.04–0.35)	0.06	(−0.14–0.25)	−0.04	(−0.17–0.08)	−0.09	(−0.28–0.10)
Adrenal and gonadal steroids ^1^												
Cortisol	−0.07	(−0.23–0.09)	−0.06	(−0.21–0.09)	0.32 **	(0.13–0.51)	0.12	(−0.11–0.36)	−0.04	(−0.19–0.12)	−0.01	(−0.25–0.23)
20αDHF	−0.03	(−0.13–0.06)	−0.05	(−0.23–0.14)	0.22 **	(0.12–0.33)	0.06	(−0.08–0.20)	−0.01	(−0.11–0.08)	−0.03	(−0.17–0.11)
20βDHF	−0.04	(−0.13–0.05)	−0.05	(−0.16–0.06)	0.20 **	(0.09–0.31)	0.07	(−0.06–0.21)	0.00	(−0.09–0.09)	−0.02	(−0.15–0.12)
20αDHE	−0.04	(−0.13–0.06)	−0.07	(−0.18–0.03)	0.22 **	(0.11–0.32)	0.08	(−0.05–0.22)	0.00	(−0.09–0.10)	−0.04	(−0.18–0.10)
20βDHE	−0.06	(−0.14–0.03)	−0.08	(−0.19–0.03)	0.16 **	(0.05–0.26)	0.04	(−0.09–0.17)	0.03	(−0.06–0.11)	−0.03	(−0.15–0.10)
Cortolone	0.02	(−0.06–0.10)	−0.02	(−0.12–0.08)	0.15 **	(0.06–0.24)	0.11	(−0.00–0.23)	0.01	(−0.07–0.08)	0.05	(−0.06–0.17)
11-dehydrocorticosterone (A)	0.04	(−0.07–0.14)	−0.04	(−0.13–0.05)	0.19 **	(0.06–0.31)	0.14	(−0.01–0.29)	0.03	(−0.08–0.13)	−0.03	(−0.18–0.13)
Testosterone	−0.03	(−0.12–0.05)	−0.04	(−0.16–0.08)	0.05	(−0.06–0.15)	0.03	(−0.09–0.15)	0.09 *	(0.01–0.17)	0.02	(−0.11–0.14)
Androstenedione (AED)	0.02	(−0.30–0.34)	0.02	(−0.08–0.11)	0.04	(−0.35–0.43)	0.10	(−0.37–0.57)	−0.12	(−0.44–0.19)	0.47 *	(0.00–0.93)
%Cortisone	−0.02	(−0.13–0.09)	0.01	(−0.03–0.06)	0.14 *	(0.00–0.27)	0.08	(−0.09–0.24)	−0.04	(−0.15–0.07)	0.02	(−0.14–0.19)
3b. Women. N = 130												
PSS	0.06	(−0.06–0.18)	2.07 **	(0.80–3.34)	−0.05	(−0.17–0.06)	−0.03	(−0.22–0.16)	−0.07	(−0.17–0.03)	−0.06	(−0.21–0.08)
Adrenal and gonadal steroids ^1^												
Cortisol	0.14 *	(0.00–0.28)	0.04	(−0.08–0.17)	0.00	(−0.13–0.14)	0.04	(−0.19–0.27)	−0.04	(−0.16–0.08)	0.02	(−0.15–0.19)
20βDHF	0.12 *	(0.02–0.21)	0.07	(−0.02–0.16)	0.03	(−0.06–0.13)	0.09	(−0.06–0.25)	−0.00	(−0.08–0.08)	0.05	(−0.06–0.17)
20αDHE	0.10 *	(0.01–0.18)	0.06	(−0.04–0.16)	0.03	(−0.06–0.11)	0.09	(−0.05–0.23)	0.02	(−0.06–0.09)	0.06	(−0.05–0.16)
Cortolone	0.11 *	(0.02–0.21)	−0.02	(−0.11–0.07)	0.08	(−0.01–0.17)	0.20 *	(0.05–0.35)	−0.01	(−0.09–0.07)	0.07	(−0.05–0.18)
20αDHE/20βDHE	−0.08	(−0.19–0.03)	−0.01	(−0.05–0.03)	−0.10 *	(−0.21–−0.00)	−0.20 *	(−0.37–−0.02)	0.09	(−0.01–0.18)	−0.02	(−0.16–0.11)
Cotisone/11-dehydrocorticosterone (E/A)	0.02	(−0.06–0.11)	0.03	(−0.08–0.14)	−0.03	(−0.11–0.06)	−0.03	(−0.18–0.11)	−0.14 **	(−0.22–−0.07)	−0.10	(−0.21–0.00)
Cortisol/Cortisone	−0.01	(−0.07–0.06)	−0.01	(−0.10–0.08)	−0.04	(−0.10–0.02)	−0.07	(−0.17–0.04)	−0.09 **	(−0.14–−0.04)	−0.09 *	(−0.16–−0.01)
%Cortisol	−0.03	(−0.07–0.00)	−0.01	(−0.08–0.05)	−0.02	(−0.05–0.02)	−0.03	(−0.09–0.02)	0.05 **	(0.02–0.08)	0.02	(−0.03–0.06)
%20βDHF	0.05 *	(0.00–0.09)	0.02	(−0.03–0.06)	0.04 *	(0.00–0.09)	0.06	(−0.02–0.13)	−0.02	(−0.06–0.02)	0.03	(−0.03–0.08)
%20αDHE	0.03	(−0.01–0.07)	0.00	(−0.05–0.05)	0.04 *	(0.00–0.07)	0.05	(−0.00–0.11)	−0.00	(−0.04–0.03)	0.03	(−0.01–0.07)
20αDHF/Cortisol	0.00	(−0.08–0.08)	−0.00	(−0.10–0.10)	0.06	(−0.02–0.13)	0.07	(−0.06–0.20)	0.07 **	(0.01–0.14)	0.08	(−0.01–0.18)
20βDHF/Cortisol	0.00	(−0.08–0.08)	0.03	(−0.05–0.11)	0.06	(−0.02–0.13)	0.07	(−0.06–0.20)	0.07 **	(0.01–0.14)	0.08	(−0.01–0.18)

^1^ All outcomes have been transformed into logarithms to correct skewness. ** *p* < 0.01, * *p* < 0.05.

## Data Availability

The data presented in this study are available on request from the corresponding author. The data are not publicly available due to their containing information that could compromise the privacy of research participants.

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
