# Peer review of "Association between Precarious Employment and Chronic Stress: Effect of Gender, Stress Measurement and Precariousness Dimensions—A Cross-Sectional Study"

_ijerph, 2022, doi:10.3390/ijerph19159099_

Round 1
Reviewer 1 Report
Please see attached.

Reviewer 2 Report
This cross-sectional study aims to examine the association between precarious employment measured by the multidimensional Employment Precariousness Scale (EPRES) and chronic stress measured by both self-perceived scales and biological hormone markers. The study also aims to determine whether there are gender differences between these associations.
Introduction: the introduction is comprehensive, the current research is well synthesized and objectives are clear. However, it is not entirely clear why “flexibility of employment conditions and labour market” (line 40) has increased the precariousness of employment – this could use more elaboration.
Methods: Methods are comprehensive and well done. It is best to provide information on the reliability and validity of the scales used in the main manuscript in addition to the protocol.
Results: The results are comprehensive, however it can use some more clarity. The occasional mention of “significant association” (i.e. line 18 after the table) implies that there are non-significant associations being discussed in these paragraphs, and it is not clear without referral to the table which of these discussed associations are significant. Additionally, there should be consistency in ordering when discussing association between explanatory and response variables. For example, line 3 after the table reads “the PSS was associated with the ‘vulnerability’ dimension”, but the preceding paragraphs reads “Precarious employment was positively associated with the PSS”.
Discussion: There is very interesting but brief mention of the stereotype of “male breadwinner” (line 100), which can use more expansion. What exactly does this stereotype consist of, and how exactly is this stereotype linked to the positive association between stress and precarious wages?
Limitations and strengths are very acutely identified. However, this study requires discussion on its implications to policy and practice, as well as future steps for further research.
Conclusion: The conclusion is well summarized.
General comments: there is inconsistency in page and line numbering following the insertion of table 3.
